# BYZANTINE-ROBUST LEARNING ON HETEROGENEOUS DATASETS VIA RESAMPLING

## ABSTRACT

In Byzantine robust distributed optimization, a central server wants to train a machine learning model over data distributed across multiple workers. However, a fraction of these workers may deviate from the prescribed algorithm and send arbitrary messages to the server. While this problem has received significant attention recently, most current defenses assume that the workers have identical data. For realistic cases when the data across workers are heterogeneous (non-iid), we design new attacks which circumvent these defenses leading to significant loss of performance. We then propose a simple resampling scheme that adapts existing robust algorithms to heterogeneous datasets at a negligible computational cost. We theoretically and experimentally validate our approach, showing that combining resampling with existing robust algorithms is effective against challenging attacks.

## 1 INTRODUCTION

Distributed or federated machine learning, where the data is distributed across multiple workers, has become an increasingly important learning paradigm both due to growing sizes of datasets, as well as privacy and security concerns. In such a setting, the workers collaborate to train a single model without transmitting their data directly over the networks (McMahan et al., 2016; Bonawitz et al., 2019; Kairouz et al., 2019). Due to the presence of either actively malicious agents in the network, or simply due to system and network failures, some workers may disobey the protocols and send arbitrary messages; such workers are also known as *Byzantine* workers (Lamport et al., 2019). Byzantine robust optimization algorithms combine the gradients received by all workers using robust aggregation rules, to ensure that the training is not impacted by the malicious workers.

While this problem has received significant recent attention, (Alistarh et al., 2018; Blanchard et al., 2017; Yin et al., 2018a), most of the current approaches assume that the data present on each different worker has identical distribution. In this work, we show that existing Byzantine-robust methods catastrophically fail in the realistic setting when the data is distributed heterogeneously across the workers. We then propose a simple resampling scheme which can be readily combined with existing aggregation rules to allow robust training on heterogeneous data.

**Contribution.** Concretely, our contributions in this work are

- We show that when the data across workers is heterogeneous, existing robust rules might not converge, even without any Byzantine adversaries.
- We propose two new attacks, normalized gradient and mimic, which take advantage of data heterogeneity and circumvent median and sign-based defenses (Blanchard et al., 2017; Pillutla et al., 2019; Li et al., 2019).
- We propose a simple new resampling step which can be used before any existing robust aggregation rule. We instantiate our scheme with KRUM and theoretically prove that the resampling generalizes it to the setting of heterogeneous data.
- Our experiments evaluate the proposed resampling scheme against known and new attacks and show that it drastically improves the performance of 3 existing schemes on realistic heterogeneously distributed datasets.

**Setup and notations.** We study the general distributed optimization problem

$$\mathcal{L}^\star = \min_{\boldsymbol{x} \in \mathbb{R}^d} \{ \mathcal{L}(\boldsymbol{x}) := \tfrac{1}{n} \sum_{i=1}^n \mathcal{L}_i(\boldsymbol{x}) \} \qquad (1)$$

where $\mathcal{L}_i : \mathbb{R}^d \to \mathbb{R}$ are the individual loss functions distributed among $n$ workers, each having its own (heterogeneous) data distribution $\{\mathcal{D}_i\}_{i=1}^n$. The case of empirical risk minimization with $m_i$ datapoints $\boldsymbol{\xi}_i \sim \mathcal{D}_i$ on worker $i$ is obtained when using $\mathcal{L}_i(\boldsymbol{x}) := \frac{1}{m_i} \sum_{j=1}^{m_i} \mathcal{L}_i(\boldsymbol{x}, \boldsymbol{\xi}_i^j)$. The (stochastic) gradient computed by a **good** node $i$ with sample $j$ is given as $\boldsymbol{g}_i(\boldsymbol{x}) := \nabla \mathcal{L}_i(\boldsymbol{x}, \boldsymbol{\xi}_i^j)$ with mean $\boldsymbol{\mu}_i$ and variance $\sigma_i^2$. We also assume that the heterogeneity (variance across good workers) is bounded i.e.

$$\mathbb{E}_i \|\nabla \mathcal{L}_i(\boldsymbol{x}) - \nabla \mathcal{L}(\boldsymbol{x})\|^2 \leq \bar{\sigma}^2 \,, \forall \boldsymbol{x} \,.$$

We write $\boldsymbol{g}_i$ instead of $\boldsymbol{g}_i(\boldsymbol{x}^t)$ when there is no ambiguity. A distributed training step using an aggregation rule is given as

$$\boldsymbol{x}^{t+1} := \boldsymbol{x}^t - \gamma^t \mathrm{Aggr}(\{\boldsymbol{g}_i(\boldsymbol{x}^t) : i \in [n]\}) \tag{2}$$

If the aggregation rule is the arithmetic mean, then (2) recovers standard minibatch SGD.

**Byzantine attack model.** In each iteration, there is a set **Byz** of at most $f$ Byzantine workers. The remaining workers are **good**, thus follow the described protocol. A Byzantine worker $j \in$ **Byz** can deviate from protocol and send an arbitrary vecter to the server. Besides, we also allow that Byzantine workers can collude with each other and know every state of the system. Unlike martingale-based approaches like (Alistarh et al., 2018), we allow the set **Byz** to change over time (Blanchard et al., 2017; Chen et al., 2017; Mhamdi et al., 2018).

## 2 RELATED WORK

There has been significant recent work of the case when the workers have identical data distributions (Blanchard et al., 2017; Chen et al., 2017; Mhamdi et al., 2018; Alistarh et al., 2018; Mhamdi et al., 2018; Yin et al., 2018a;b; Su & Xu, 2018; Damaskinos et al., 2019). We discuss the most pertinent of these methods next. Blanchard et al. (2017) formalize the Byzantine robust setup and propose a distance-based approach KRUM which selects a worker whose gradient is very close to at least half the other workers. A different approach involves using the median and its variants (Blanchard et al., 2017; Pillutla et al., 2019; Yin et al., 2018a). Yin et al. (2018a) propose to use and analyze the coordinate-wise median method (CM). Pillutla et al. (2019) use a smoothed version of Weiszfeld's algorithm to iteratively compute an approximate geometric median of the input gradients. In a third approach, (Bernstein et al., 2018) propose to use the signs of the gradients and then aggregate them by majority vote, however, (Karimireddy et al., 2019) show that it may not always converge. Finally, Alistarh et al. (2018) use a martingale-based aggregation rule which gives a sample complexity optimal algorithm for iid data. The distance-based approach of KRUM was later extended in Mhamdi et al. (2018) who propose BULYAN to overcome the *dimensional leeway* attack. This is the so called *strong Byzantine resilience* and is orthogonal to the question of non-iid-ness we study here. Recently, (Peng & Ling, 2020; Yang & Bajwa, 2019a;b) studied Byzantine-resilient algorithms in the decentralized setting where there is no central server available. Extending our techniques to the decentralized setting is an important direction for future work.

In a different line of work, (Lai et al., 2016; Diakonikolas et al., 2019) develop sophisticated spectral techniques to robust estimate the mean of a high dimensional multi-variate standard Gaussian distribution where samples are evenly distributed in all directions and the attackers are concentrated in one direction. Very recent work (Data & Diggavi, 2020) extend the theoretical analysis to non-convex, strongly-convex and non-i.i.d setup under a gradient dissimilarity assumption and propose a gradient compression scheme on top of it. Our resampling trick can be combined with it to further reduce gradient dissimilarity.

Many attacks have been devised for distributed training. For the iid setting, the state-of-the-art attacks are (Baruch et al., 2019; Xie et al., 2019b). The latter attack is very strong when the fraction of adversaries is large (nearly half), but in this work we focus on settings when this fraction is quite small (e.g. $\leq 0.2$). Further our normalized mean attack Section 3.2 is inspired by (Xie et al., 2019b). The former work focuses on attacks which are coordinated across time steps. Developing strong practical defenses even in the iid case against such time-coordinated attacks remains an open problem. In this work, we sidestep this issue by restricting ourselves to new attacks made possible by non-iid data and studying how to overcome them. We focus on schemes which work in the iid setting, but fail with non-iid data. Once a new method which can defend against (Baruch et al., 2019) is developed, our proposed scheme shows how to adapt such a method to the important non-iid case. For the non-iid setting, backdoor attacks are designed to take advantage of heavy-tailed data and manipulate

model inference on specific subtask, rather than lower the overall accuracies of training (Bagdasaryan et al., 2018; Bhagoji et al., 2018). In contrast, this paper is not intended to address aforementioned challenges but rather to defend the attacks that lower the training accuracies in the non-iid setting.

As far as we are aware, only (Li et al., 2019; Ghosh et al., 2019; Sattler et al., 2020) explicitly investigate Byzantine robustness with non-iid workers. Li et al. (2019) proposes an SGD variant (RSA) which modifies the original objective by adding an $\ell_1$ penalty. Ghosh et al. (2019); Sattler et al. (2020) assume that all workers belong to an apriori fixed number of clusters and use an outlier-robust clustering method to recover these clusters. If we assume that the server has the entire training dataset and can control the distribution of samples to good workers, Xie et al. (2019a); Chen et al. (2018); Rajput et al. (2019) show that non-iid-ness can be overcome. Typical examples of this is distributed training of neural networks on public cloud, or volunteer computing Meeds et al. (2015); Miura & Harada (2015). However, none of these methods are applicable in the standard federated learning setup we consider here. We aim to minimize the original loss function over workers while respecting the non-iid data locality, i.e. the partition of the given heterogeneous dataset over the workers, without data transfer.

## 3 ATTACKS AGAINST EXISTING AGGREGATION SCHEMES

In this section we show that when the data across the workers is heterogeneous (non-iid), then we can design new attacks which take advantage of the heterogeneity, leading to the failure of existing aggregation schemes. We study three classes of robust aggregation schemes: i) schemes which select a representative worker in each round (e.g. KRUM (Blanchard et al., 2017)), ii) schemes which use normalized means (e.g. RSA (Li et al., 2019)), and iii) those which use the median (e.g. RFA (Pillutla et al., 2019)). We show realistic settings under which each of these classes would fail when faced with heterogeneous data.

### 3.1 FAILURE OF REPRESENTATIVE WORKER SCHEMES ON NON-IID DATA

Algorithms like KRUM select workers who are representative of a majority of the workers, by relying on statistics such as pairwise differences between the various worker updates. Let $(\boldsymbol{g}_1, \ldots, \boldsymbol{g}_n)$ be the gradients by the workers, $f$ of which are Byzantine (e.g. $n \geq 2f + 3$ for KRUM). For $i \neq j$, let $i \to j$ denote that $\boldsymbol{g}_j$ belongs to the $n - f - 2$ closest vectors to $\boldsymbol{g}_i$. Then KRUM is defined as follows

$$\text{KRUM}(\boldsymbol{g}_1, \ldots, \boldsymbol{g}_n) := \arg\min_i \sum_{i \to j} \|\boldsymbol{g}_i - \boldsymbol{g}_j\|^2 \qquad (3)$$

However, when the data across the workers is heterogeneous, there is no 'representative' worker. This is because each worker computes their local gradient over vastly different local data. Hence, for convergence it is important to not only select a good (non-Byzantine) worker, but also ensure that each of the good workers is selected with roughly equal frequency. Hence KRUM suffers a significant loss in performance with heterogeneous data, even when there are *no Byzantine workers*.

For example, when KRUM is used for iid datasets without adversary ($f = 0$, see left of Figure 1a), the test accuracy is close to simple average and the gap can be filled by MULTI-KRUM (Blanchard et al., 2017). The right plot of Figure 1a also shows that KRUM's selection of gradients is biased towards certain nodes. When KRUM is applied to non-iid datasets (the middle of Figure 1a), KRUM performs poorly even without any attack. This is because KRUM mostly selects gradients from a few nodes whose distribution is closer to others (the right of Figure 1a). This is an example of how robust aggregation rules may fail on realistic non-iid datasets.

### 3.2 ATTACKS ON NORMALIZED AGGREGATION SCHEMES

Instead of simply averaging the gradients, some methods first normalize them and then average. This limits the influence of the Byzantine workers since they cannot output extremely large gradients, and hence is more robust. For example RFA (Pillutla et al., 2019) with $T=1$ uses following aggregation rule:

$$\text{NM}(\boldsymbol{g}_1, \ldots, \boldsymbol{g}_n) = \sum_{i=1}^n \frac{\boldsymbol{g}_i}{\|\boldsymbol{g}_i\|_2} \qquad (4)$$

Other methods such as RSA (Li et al., 2019) or signum (Bernstein et al., 2018) normalize entries coordinate-wise before taking a majority vote i.e. update the server model $\boldsymbol{x}_0$ on server using local model $\boldsymbol{x}_i$ from node $i$ (not gradient) using

$$\text{RSA}(\boldsymbol{x}_0; \boldsymbol{x}_1, \ldots, \boldsymbol{x}_n) := \nabla f_0(\boldsymbol{x}_0) + \lambda \sum_{i=1}^n \text{sign}(\boldsymbol{x}_0 - \boldsymbol{x}_i) \qquad (5)$$

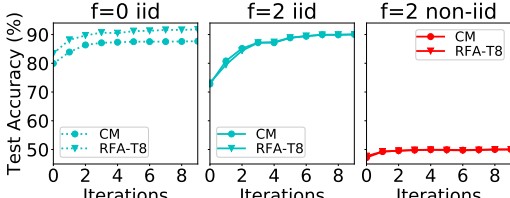

(a) Left & middle: Comparing arithmetic mean with KRUM on iid and non-iid datasets, **without** any Byzantine workers. Right: Histogram of selected gradients.

(b) Comparing normalized mean (RFA with T=1) under the **normalized mean attack** with $f = 0, 1, 2$ attackers.

(c) Comparing coordinate-wise median (CM) and geometric median (RFA with T=8) under the **mimic2 attack** on iid and non-iid datasets.

Figure 1: Failures of existing aggregation rules on the non-iid MNIST dataset. In all experiments, there are 8 good and $f$ Byzantine workers.

where $f_0$ is a strongly convex penalty term and $\lambda > 0$ is a relaxation parameter.

However, a Byzantine worker can still craft an "omniscient" attack to foil robust aggregations, using an approach similar to the negative sum for the arithmetic mean (Blanchard et al., 2017; Li et al., 2019):

$$\boldsymbol{v} := - \textstyle\sum_{i \in \mathbf{good}} \frac{\boldsymbol{g}_i}{\|\boldsymbol{g}_i\|_2} \tag{6}$$

On the right side of Figure 1b, we can see that this attack lowers the accuracy of RFA-T1 significantly, as the number of Byzantine workers increases. Comparing to its iid counterpart, the normalized mean attack is even more impactful in the non-iid setting.

### 3.3 ATTACKS ON MEDIAN-BASED SCHEMES

Geometric median and its variants are popular in robust learning research (Blanchard et al., 2017; Chen et al., 2017; Pillutla et al., 2019; Yin et al., 2018a; Mhamdi et al., 2018). Given gradients $\{\boldsymbol{g}_1, \ldots, \boldsymbol{g}_n\}$, we use the estimator

$$\mathrm{GM}(\boldsymbol{g}_1, \ldots, \boldsymbol{g}_n) := \mathrm{argmin}_{\boldsymbol{v}} \textstyle\sum_{i=1}^{n} \|\boldsymbol{v} - \boldsymbol{g}_i\|. \tag{7}$$

If the vectors $\{\boldsymbol{g}_1, \ldots, \boldsymbol{g}_n\}$ are drawn independently from the same distribution, intuitively most of them would concentrate around their mean. Then, even if there are some Byzantine outputs, the median would ignore those as outliers and output a 'central' point close to the mean.

However, when $\{\boldsymbol{g}_1, \ldots, \boldsymbol{g}_n\}$ are gradients over heterogeneous data, they may be vastly different from each other and do not concentrate around the mean. In such a scenario, the median such as (7) can be even less robust than simply taking the mean. Suppose that worker 0 is Byzantine and the remaining workers $\{1, \ldots, 2n\}$ are good, with a total of $2n + 1$ workers. Now suppose that $g_i = (-1)^i$ for all the workers, with half the good workers having $-1$ and the other half $+1$. This means that the true mean is 0, however, the median estimator (7) will output 1.

**Mimic attack.** This motivates our *mimic* attack in which all Byzantine workers collude and agree to always send gradients from the same worker. We define a specialized attack, called *mimic2*, where half of the good workers have same datasets and send $\boldsymbol{g}_1$ while the rest good workers send $\boldsymbol{g}_2$; then all Byzantine workers send $\boldsymbol{v} = \boldsymbol{g}_1$ such that the geometric median of the gradients received by the server is always $\boldsymbol{g}_1$. Therefore, this attack breaks geometric-median-based robust aggregation rules, by leading them to wrong solutions. The left plot of Figure 1c shows the impact of the mimic2 attack. Test accuracies of CM and RFA both drop drastically to around 50%.

---

**Algorithm 1** Robust Learning with Resampling

---

**Setup**: $n$ workers, $f$ of which are Byzantine; resampling $T$ times, each time samples $s$ gradients. A robust learning algorithm AGGR on iid datasets; $\gamma$ is the learning rate.

**Workers**:
1. Each good worker $i$ randomly samples a datapoint $j$ and computes a stochastic gradient $\boldsymbol{g}_i := \nabla F_i(\boldsymbol{x}, \boldsymbol{\xi}_i^j)$ where $\boldsymbol{\xi}_i^j \sim \mathcal{D}_i$; each Byzantine worker $i$ sends arbitrary vector $\boldsymbol{g}_i$.
2. Send $\boldsymbol{g}_i$ to server.

**Servers**:
1. Receive $\{\boldsymbol{g}_i\}_{i=1}^n$ from all workers.
2. $\mathcal{S}, \mathcal{I}_S$ = Resampling($\{\boldsymbol{g}_i : i \in [n]\}, f, T, s$); See Algorithm 2 .
3. Compute $\boldsymbol{x}' := \boldsymbol{x} - \gamma \text{AGGR}(\mathcal{S})$;
4. Broadcast $\boldsymbol{x}'$ to all workers.

---

**Algorithm 2** Resampling with $s$-replacement

---

**Input:** $\{\boldsymbol{g}_i : i \in [n]\}, T := n, s, \{c[i] := 0 : i \in [n]\}$
**for** $t := 1, \ldots, T$ **do**
   **for** $i := 1, \ldots, s$ **do**
      **while** Select $j_i \sim \textbf{Uniform}([n])$ **do**
         **if** $c[j_i] < s$ **then**
            $c[j_i] += 1$
            If $c[j_i] == s$ Break;
   Compute average $\bar{\boldsymbol{g}}_t := \frac{1}{s}\sum_{i=1}^s \boldsymbol{g}_{j_i}$
Return $\{\bar{\boldsymbol{g}}_t : t \in [T]\}, \{j_i^t : t \in [T], i \in [s]\}$

---

## 4    ROBUST AGGREGATION ON NON-IID DATA

In Section 3 we have demonstrated how existing robust aggregation rules can fail in realistic non-iid scenarios, with and without attackers (Sections 3.2 and 3.3 and Section 3.1 respectively). To overcome this problem, we propose a simple new resampling-based aggregation rule for training, shown in Algorithm 1. More specifically, we choose *s-resampling without replacement* in Algorithm 2 where each gradient can be sampled at most $s$ times. The key property of our rule is that after resampling, the resulting set of averaged gradients $\{\bar{\boldsymbol{g}}_t : t \in [T]\}$ are much more homogeneous (lower variance). Then these averaged gradients are fed to existing Byzantine robust aggregation schemes, such as KRUM, see Section 5. Given an existing aggregation rule AGGR, we denote by AGGR $\circ$ Resampling the resulting new robust aggregation rule for non-iid input gradients.

In the following proposition, we list the desired properties of Algorithm 2

**Proposition I.** *Given a population $\{\boldsymbol{g}_i : i \in [n]\} \subset \mathbb{R}^d$ of mean $\boldsymbol{\mu} := \frac{1}{n}\sum_{i=1}^n \boldsymbol{g}_i$ and variance $\sigma^2 := \frac{1}{n}\sum_{i=1}^n \|\boldsymbol{g}_i - \boldsymbol{\mu}\|^2$, let $\{\bar{\boldsymbol{g}}_t : t \in [T]\}$ be the output of Algorithm 2 on $\{\boldsymbol{g}_i : i \in [n]\}$. Then*

- *If there are no Byzantine workers, then $\{\bar{\boldsymbol{g}}_t : t \in [T]\}$ are identically distributed*
$$\mathbb{E}[\bar{\boldsymbol{g}}_t] = \boldsymbol{\mu}, \quad var(\bar{\boldsymbol{g}}_t) = \frac{n-1}{sn-1}\sigma^2 \qquad \forall \, t \in [T] \tag{8}$$

- *If $f$ of the $n$ inputs are Byzantine, then at least $T - sf$ gradients in $\{\bar{\boldsymbol{g}}_t : t \in [T]\}$ are good; that is, a good $\bar{\boldsymbol{g}}_t$ is the average of gradients $\{\boldsymbol{g}_{j_i^t} : i \in [s]\} \subset \boldsymbol{good} \subset [n]$. Then such good $\{\bar{\boldsymbol{g}}_t\}$ are identically distributed with*
$$\mathbb{E}[\bar{\boldsymbol{g}}_t] = \tilde{\boldsymbol{\mu}}, \quad var(\bar{\boldsymbol{g}}_t) = \frac{n-1}{sn-1}\tilde{\sigma}^2 \tag{9}$$
*where $\tilde{\boldsymbol{\mu}} := \frac{1}{|\boldsymbol{good}|}\sum_{i \in \boldsymbol{good}} \boldsymbol{g}_i$, and $\tilde{\sigma}^2 := \frac{1}{|\boldsymbol{good}|}\sum_{i \in \boldsymbol{good}} \|\boldsymbol{g}_i - \mathbb{E}[\bar{\boldsymbol{g}}_t]\|^2$.*

*Proof.* Since Algorithm 2 resamples $s$ gradients to estimate a population of $sn$ samples, we can use sampling theory (Middleton, 1988, Ch. Survey Sampling) to compute the sample mean
$$\mathbb{E}\big[\textbf{RS}(\boldsymbol{g}_1, \ldots, \boldsymbol{g}_n)\big] = \boldsymbol{\mu} \tag{10}$$
and the sample variance
$$\mathbb{E}\big[(\textbf{RS}(\boldsymbol{g}_1, \ldots, \boldsymbol{g}_n) - \boldsymbol{\mu})^2\big] = \frac{1}{s}\big(1 - \frac{s-1}{sn-1}\big)\sigma^2 = \frac{n-1}{sn-1}\sigma^2. \tag{11}$$
Since the gradients are sampled at most $s$ times, at most $sf$ out of the $T$ gradients are affected by a Byzantine worker. Its mean and variance can be calculated in the same way shown above. $\qquad \square$

**Remark 1.** *For $s = 1$, resampling simply becomes shuffling of the input elements, and $var(\bar{g}_t) = \sigma^2$ is unchanged. For $s > 1$, the resampling scheme reduces the heterogeneity (variance) by approximately $1/s$. Thus, increasing $s$ leads to the resulting resampled gradients being a better estimator of the population mean, thus improving training convergence speed. On the other hand, increasing $s$ also increases the number of resampled gradients which can be affected by a Byzantine worker. In particular, if $f$ workers are Byzantine, then up to $fs$ resampled gradients can be incorrect, which has to be taken into account by the employed robust aggregation rule. In practice, we found that using a small value $s = 2$ was already sufficient to overcome heterogeneity.*

**Remark 2.** *A natural question to ask is what happens if we resample with replacement but do not limit on the number of replacements. We discuss this additional algorithm variant in Appendix C.*

Note that the $\{\bar{g}_t : t \in [T]\}$ are identically distributed but not independent. This does not directly fit into the original assumptions of Byzantine robust algorithms like KRUM and hence the robustness has to be reproved for our more general setting.

## 5 CONVERGENCE ANALYSIS WITH KRUM

In this section, we analyze the convergence of SGD with robust aggregation on non-iid data. Since the definition of robustness and other conditions vary from paper to paper, it is not possible to give a uniform proof perfectly fit for all methods. For example, (Yin et al., 2018a) assumes the gradients have bounded variance and skewness whereas others like KRUM, RFA, BULYAN does not. Thus we only analyze KRUM for its simplicity and popularity, and show that analysis is only slightly different from the original version. For other algorithms, we show by experiments that resampling helps them achieve better performance on heterogeneous data, see Section 6.

Definition A generalizes the Byzantine resilience of (Blanchard et al., 2017, Definition 1) to the cases where we have non-iid data. Let $G$ be an estimator of the good gradients.

**Definition A** (($\alpha, f$)-Byzantine Resilience.)**.** *Let $0 \leq \alpha < \pi/2$ be any angular value, and any integer $0 \leq f \leq n$. Let $\mathcal{B} = \{j_1, \ldots, j_f : j_1 \leq j_1 < \cdots < j_f \leq n\}$ be the indices of Byzantine workers. Let $\{V_i \in \mathcal{D}_i : i \in [n]\backslash\mathcal{B}\}$ be independent random vectors in $\mathbb{R}^d$. Let $G = G(\boldsymbol{\xi})$ be an independent random variable which randomly selects a good worker $i$ and samples a vector from $\mathcal{D}_i$ and $\mathbb{E}\, G = \boldsymbol{g}$. Let $B_1, \ldots, B_f$ be any Byzantine vectors in $\mathbb{R}^d$, possibly dependent on the $V_i$'s. An aggregation rule $F$ is said to be ($\alpha, f$)-Byzantine resilient if*

$$F = F(V_1, \ldots, \underbrace{B_1}_{j_1}, \ldots, \underbrace{B_f}_{j_f}, \ldots V_n)$$

*satisfies (i) $\langle \mathbb{E}\, F, \boldsymbol{g} \rangle \geq (1 - \sin \alpha) \cdot \|\boldsymbol{g}\|^2 > 0$ and (ii) for $r = 2, 3, 4$, $\mathbb{E}\, \|F\|^r \leq \mathbb{E}\, \|G\|^r$.*

Then we can conclude the almost sure convergence similar to (Blanchard et al., 2017, Proposition 2)

**Theorem II** (Resampling KRUM)**.** *We assume that (i) the cost function $\mathcal{L}$ is three times differentiable with continuous derivatives and non-negative $\mathcal{L}(\boldsymbol{x}) \geq 0$; (ii) the learning rates satisfy $\sum_t \gamma_t = \infty$ and $\sum_t \gamma_t^2 < \infty$. Let the good workers have stochastic gradients $G_i(\boldsymbol{x}, \boldsymbol{\xi})$ for $i \in \textbf{good} \subset [n]$. We assume that for a uniformly chosen $j \in \textbf{good}$, the following is true (iii) $\mathbb{E}_{j,\boldsymbol{\xi}}[G_j(\boldsymbol{x}, \boldsymbol{\xi})] = \nabla\mathcal{L}(\boldsymbol{x})$ and $\forall r \in \{2, 3, 4\}$, $\mathbb{E}_{j,\boldsymbol{\xi}} \|G_j(\boldsymbol{x}, \boldsymbol{\xi})\|^r \leq A_r + B_r\|\boldsymbol{x}\|^r$ for some constants $A_r, B_r$; (iv) there exists a constant $0 \leq \alpha < \pi/2$ such that for all $\boldsymbol{x}$ we have $\eta(T, sf) \cdot \sqrt{d} \cdot \sigma(\boldsymbol{x}) \leq \|\nabla\mathcal{L}(\boldsymbol{x})\| \cdot \sin \alpha$ where $\sigma^2(\boldsymbol{x}) := \frac{n-1}{sn-1}\tilde{\sigma}^2(\boldsymbol{x})$; (v) finally, beyond a certain horizon, $\|\boldsymbol{x}\|^2 \geq D$, there exist $\varepsilon > 0$ and $0 \leq \beta < \pi/2 - \alpha$ such that $\|\nabla\mathcal{L}(\boldsymbol{x})\| \geq \varepsilon > 0$ and $\langle \boldsymbol{x}, \nabla\mathcal{L}(\boldsymbol{x}) \rangle \geq \cos\beta\|\boldsymbol{x}\| \cdot \|\nabla\mathcal{L}(\boldsymbol{x})\|$. If $s > 1$ and $2sf + 3 \leq n$, then*

- *KRUM ○Resampling is ($\alpha, sf$)-Byzantine resilient where $0 \leq \alpha < \pi/2$ is defined by*

$$\sin \alpha = \frac{\eta(T, sf) \cdot \sqrt{d} \cdot \sigma}{\|\nabla\mathcal{L}(\boldsymbol{x})\|}, \quad \eta(n, f) := \sqrt{2\left(n - f + \frac{f \cdot (n-f-2) + f^2 \cdot (n-f-1)}{n-2f-2}\right)} \tag{12}$$

- *the sequence of gradients $\nabla\mathcal{L}(\boldsymbol{x}_t)$ converges almost surely to zero.*

We defer the proof to Appendix A. The above convergence result for heterogeneous data is nearly identical to (Blanchard et al., 2017, Proposition 2) for iid data, except for the slightly stronger restriction on the number of Byzantine workers $2sf + 3 \leq n$.

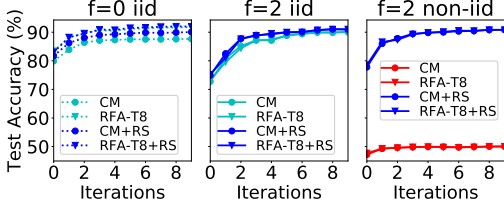

(a) Left & middle: Comparing arithmetic mean with KRUM on iid and non-iid datasets, **without** any Byzantine workers. Right: Histogram of selected gradients.

(b) Comparing normalized mean (RFA with T=1) under the **normalized mean attack** with $f = 0, 1, 2$ attackers.

(c) Comparing coordinate-wise median (CM) and geometric median (RFA with T=8) under **mimic2 attack** on iid and non-iid datasets.

Figure 2: Combining resampling with existing aggregation rules on non-iid MNIST dataset. In all experiments, there are 8 good and $f$ Byzantine workers. For each aggregation we resample and average $s$ gradients for $T = n$ times.

## 6 EXPERIMENTS

In this section, we demonstrate the effect of resampling on datasets distributed in a non-iid fashion. Throughout the section, we illustrate the challenge, attacks, and defense by an example of training an MLP on the MNIST dataset (LeCun et al., 1998). In Appendix D, we present the results of similar experiments on Fashion-MNIST (Xiao et al., 2017) and CIFAR-10 (Krizhevsky et al., 2009). The dataset is sorted by labels and sequentially divided into equal parts among good workers; Byzantine workers have access to the dataset on all good workers. Implementations are based on PyTorch (Paszke et al., 2019) and will be made publicly available.

### 6.1 RESAMPLING AGAINST THE ATTACKS ON NON-IID DATA

In Section 3 we have presented how heterogeneous data can lead to failure of existing robust aggregation rules. Here we apply our proposed resampling with $T=n$, $s=2$ to the same aggregation rules, showing that resampling overcomes the described failures. Results are presented in Figure 2. In Figure 2a, we show that using resampling helps KRUM to achieve better test accuracy on non-iid data. Since resampling KRUM with $s=2$ actually averages 2 gradients, we compare it with MULTIKRUM with $m=2$. The middle of Figure 2a shows that MULTIKRUM with $m=2$ performs better than KRUM, but KRUM with resampling is even better which suggests the resampling step improves the performance on non-iid data. The selection histogram on the rightmost part of Figure 2a shows that after resampling, KRUM's selection is much more evenly distributed between the good workers. In Figure 2b, we show that resampling fixes RFA with $T=1$ and allows it to defend against the normalized mean attack. The resampling-based aggregation can almost reach same accuracy for both iid and non-iid setup. In Figure 2c, while mimic attack does not work for median-based rules in the iid setting, resampling still slightly improves the performance due to variance reduction. In the non-iid setting, resampling drastically improves the accuracy to the same level as the iid setting.

### 6.2 RESAMPLING AGAINST GENERAL BYZANTINE ATTACKS

In Figure 3, we present thorough experiments on non-iid data over 10 workers with 2 Byzantine workers. In each subfigure, we compare an aggregation rule with its variant with resampling. Three aggregation rules are compared: KRUM, CM, RFA. In particular, we compare to RFA with both T=1 (normalized mean) and T=8 (geometric median).

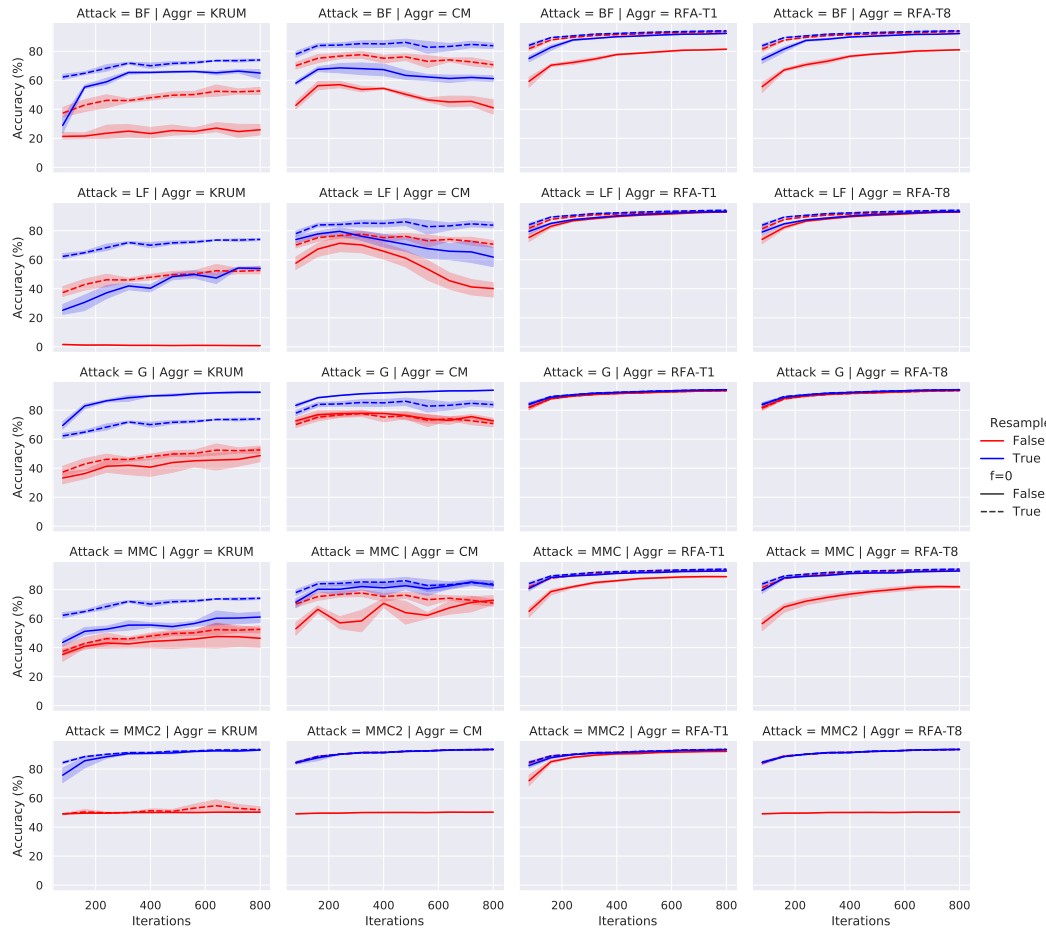

Figure 3: Test accuracies of KRUM, CM, RFA under 5 kinds of attacks (and without attack) on non-iid datasets. There are 10 workers and 2 of them are Byzantine according to each attack row. Columns show each aggregation rule applied without (red) and with resampling (blue). Dashed lines for comparison are showing the same method without any Byzantine workers ($f = 0$). For RFA, T1, T8 refers to the number of inner iterations of Weiszfeld's algorithm.

**Attacks.** 5 different kinds of attacks are applied (one per row in the figure): bitflipping, labelflipping, gaussian attack, as well as the mimic and mimic2 attacks.

- **Bitflipping**: A Byzantine worker flips the sign bits and sends $-\nabla f(\boldsymbol{x})$ instead of $\nabla f(\boldsymbol{x})$ because of problems like hardware failures etc.
- **Labelflipping**: The dataset on workers have corrupted labels. For the MNIST dataset, we let Byzantine workers transform labels by $\mathcal{T}(y) := 9 - y$.
- **Gaussian**: A Byzantine worker sends a Gaussian random vector of 0 mean and isotropic covariance matrix with standard deviation 200 (Xie et al., 2018).
- **mimic & mimic2**: Explained in Section 3.3.

From Figure 3 we can see that resampling improves the accuracy on most of the tasks. The final accuracies achieved vary with the aggregation rules we use. Notice that RFA-T1 is more robust to the mimic attack than RFA-T8 in Figure 3 because more inner iterations lead to better approximate geometric median and less robust to normalized mean attacks. The normalized mean attack has been addressed in Section 3.2.

## 7 CONCLUSION

In this paper, we initiated a study of robust distributed learning problem under realistic heterogeneous data. We showed that many existing Byzantine-robust aggregation rules fail under simple new attacks, or sometimes even without any Byzantine workers. As a solution, we propose a resampling scheme

which effectively adapts existing robust algorithms to heterogeneous datasets at a negligible computational cost. We believe robustness under heterogeneous conditions has been an overlooked direction of research thus far and hope to inspire more work on this topic. Extending to the decentralized setting, stronger Byzantine adversaries, as well as obtaining optimal algorithms are other challenging directions for future work.

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

## A   CONVERGENCE ANALYSIS OF KRUM WITH RESAMPLING

**Theorem II** (Resampling KRUM). *We assume that (i) the cost function $\mathcal{L}$ is three times differentiable with continuous derivatives and non-negative $\mathcal{L}(\boldsymbol{x}) \geq 0$; (ii) the learning rates satisfy $\sum_t \gamma_t = \infty$ and $\sum_t \gamma_t^2 < \infty$. Let the good workers have stochastic gradients $G_i(\boldsymbol{x}, \boldsymbol{\xi})$ for $i \in \textbf{good} \subset [n]$. We assume that for a uniformly chosen $j \in \textbf{good}$, the following is true (iii) $\mathbb{E}_{j,\boldsymbol{\xi}}[G_j(\boldsymbol{x}, \boldsymbol{\xi})] = \nabla \mathcal{L}(\boldsymbol{x})$ and $\forall r \in \{2, 3, 4\}$, $\mathbb{E}_{j,\boldsymbol{\xi}} \|G_j(\boldsymbol{x}, \boldsymbol{\xi})\|^r \leq A_r + B_r \|\boldsymbol{x}\|^r$ for some constants $A_r, B_r$; (iv) there exists a constant $0 \leq \alpha < \pi/2$ such that for all $\boldsymbol{x}$ we have $\eta(T, sf) \cdot \sqrt{d} \cdot \sigma(\boldsymbol{x}) \leq \|\nabla \mathcal{L}(\boldsymbol{x})\| \cdot \sin \alpha$ where $\sigma^2(\boldsymbol{x}) := \frac{n-1}{sn-1} \tilde{\sigma}^2(\boldsymbol{x})$; (v) finally, beyond a certain horizon, $\|\boldsymbol{x}\|^2 \geq D$, there exist $\varepsilon > 0$ and $0 \leq \beta < \pi/2 - \alpha$ such that $\|\nabla \mathcal{L}(\boldsymbol{x})\| \geq \varepsilon > 0$ and $\langle \boldsymbol{x}, \nabla \mathcal{L}(\boldsymbol{x}) \rangle \geq \cos \beta \|\boldsymbol{x}\| \cdot \|\nabla \mathcal{L}(\boldsymbol{x})\|$. If $s > 1$ and $2sf + 3 \leq n$, then*

- KRUM $\circ$ *Resampling is $(\alpha, sf)$-Byzantine resilient where $0 \leq \alpha < \pi/2$ is defined by*

$$\sin \alpha = \frac{\eta(T, sf) \cdot \sqrt{d} \cdot \sigma}{\|\nabla \mathcal{L}(\boldsymbol{x})\|}, \quad \eta(n, f) := \sqrt{2\big(n - f + \frac{f \cdot (n-f-2) + f^2 \cdot (n-f-1)}{n - 2f - 2}\big)} \quad (12)$$

- *the sequence of gradients $\nabla \mathcal{L}(\boldsymbol{x}_t)$ converges almost surely to zero.*

*Proof.* We only prove the first statement. The second one follows from applying (Blanchard et al., 2017, Proposition 2) directly.

After resampling, we have $\tilde{n} := T$ gradients, and at most $\tilde{f} := sf$ of them are Byzantine. Without loss of generality, we assume that Byzantine vectors $B_1, \ldots, B_{\tilde{f}}$ occupy the last $\tilde{f}$ positions in the arguments of KRUM. We denote as KRUM $:= $ KRUM$(\tilde{V}_1, \ldots, \tilde{V}_{\tilde{n}-\tilde{f}}, \tilde{B}_1, \ldots, \tilde{B}_{\tilde{f}})$. For each index $i$, we denote by $\delta_c(i)$ (resp. $\delta_b(i)$) the number of correct (resp. Byzantine) indices $j$ such that $i \to j$ (recall again that $i \to j$ denotes that $V_j$ belongs to the $n - f - 2$ closest vectors to $V_i$). We have

$$\delta_c(i) + \delta_b(i) = \tilde{n} - \tilde{f} - 2$$
$$\tilde{n} - 2\tilde{f} - 2 \leq \delta_c(i) \leq \tilde{n} - \tilde{f} - 2$$
$$\delta_b(i) \leq \tilde{f}$$

We focus first on the condition (i) of $(\alpha, \tilde{f})$-Byzantine resilience as in Definition A. We determine an upper bound on the squared distance $\|\mathbb{E}\text{KRUM} - \boldsymbol{g}\|^2$. Note that, for any correct $j$, $\mathbb{E}\tilde{V}_j = \boldsymbol{g}$. Let $i_*$ be the index chosen by KRUM.

$$\|\mathbb{E}\text{KRUM} - \boldsymbol{g}\|^2 \leq \|\mathbb{E}(\text{KRUM} - \tfrac{1}{\delta_c(i_*)} \sum_{i_* \to \text{correct}j} \tilde{V}_j)\|^2$$
$$\leq \mathbb{E}\|\text{KRUM} - \tfrac{1}{\delta_c(i_*)} \sum_{i_* \to \text{correct}j} \tilde{V}_j\|^2$$
$$\leq \sum_{\text{correct } i} \mathbb{E}\|\tilde{V}_i - \tfrac{1}{\delta_c(i)} \sum_{i \to \text{correct}j} \tilde{V}_j\|^2 \mathbb{1}(i_* = i)$$
$$+ \sum_{\text{byz } k} \mathbb{E}\|\tilde{B}_k - \tfrac{1}{\delta_c(k)} \sum_{k \to \text{correct}j} \tilde{V}_j\|^2 \mathbb{1}(i_* = k)$$

where $\mathbb{1}$ is the indicator function. Examine $i_* = i$ for correct index $i$.

$$\|\tilde{V}_i - \tfrac{1}{\delta_c(i)} \sum_{i \to \text{correct}j} \tilde{V}_j\|^2 = \|\tfrac{1}{\delta_c(i)} \sum_{i \to \text{correct}j} \tilde{V}_i - \tilde{V}_j\|^2$$
$$\leq \tfrac{1}{\delta_c(i)} \sum_{i \to \text{correct}j} \|\tilde{V}_i - \tilde{V}_j\|^2$$
$$\mathbb{E}\|\tilde{V}_i - \tfrac{1}{\delta_c(i)} \sum_{i \to \text{correct}j} \tilde{V}_j\|^2 \leq \tfrac{1}{\delta_c(i)} \sum_{i \to \text{correct}j} \mathbb{E}\|\tilde{V}_i - \tilde{V}_j\|^2$$
$$\leq 2d\sigma^2$$

where $\sigma^2 := \frac{n-1}{sn-1} \tilde{\sigma}^2$ in Proposition I. We now examine the case $i_* = k$ for some Byzantine index $k$. The fact that $k$ minimizes the score implies for all correct $i$

$$\sum_{k \to \text{correct}j} \|\tilde{B}_k - \tilde{V}_j\|^2 + \sum_{k \to \text{byz } l} \|\tilde{B}_k - \tilde{B}_l\|^2 \leq \sum_{i \to \text{correct}j} \|\tilde{V}_i - \tilde{V}_j\|^2 + \sum_{i \to \text{byz } l} \|\tilde{V}_i - \tilde{B}_l\|^2 \tag{13}$$

Then, for all correct indices $i$

$$\|\tilde{B}_k - \tfrac{1}{\delta_c(k)} \sum_{k \to \text{correct}j} \tilde{V}_j\|^2 \leq \tfrac{1}{\delta_c(k)} \sum_{k \to \text{correct}j} \|\tilde{B}_k - \tilde{V}_j\|^2$$
$$\leq \tfrac{1}{\delta_c(k)} \sum_{i \to \text{correct}j} \|\tilde{V}_i - \tilde{V}_j\|^2 + \tfrac{1}{\delta_c(k)} \sum_{i \to \text{byz } l} \|\tilde{V}_i - \tilde{B}_l\|^2$$

Focus on $D^2(i) := \sum_{i \to \text{byz } l} \|\tilde{V}_i - \tilde{B}_l\|^2$, each correct worker $i$ has $\tilde{n} - \tilde{f} - 2$ neighbors, and $\tilde{f} + 1$ non-neighbors. Thus there exists a correct $\zeta(i)$ which is farther from $i$ than any of the neighbors of $i$. In particular, for each Byzantine index $l$ such that $i \to l$, $\|\tilde{V}_i - \tilde{B}_l\|^2 \le \|\tilde{V}_i - \tilde{V}_{\zeta(i)}\|^2$. Whence

$$\|\tilde{B}_k - \tfrac{1}{\delta_c(k)}\sum_{k\to\text{correct}j}\tilde{V}_j\|^2 \le \tfrac{1}{\delta_c(k)}\sum_{i\to\text{correct}j}\|\tilde{V}_i - \tilde{V}_j\|^2 + \tfrac{\delta_b(i)}{\delta_c(k)}\|\tilde{V}_i - \tilde{V}_{\zeta(i)}\|^2$$

$$\mathbb{E}\|\tilde{B}_k - \tfrac{1}{\delta_c(k)}\sum_{k\to\text{correct}j}\tilde{V}_j\|^2 \le \tfrac{\delta_c(i)}{\delta_c(k)}2d\sigma^2 + \tfrac{\delta_b(i)}{\delta_c(k)}\sum_{\text{correct } j\neq i}\mathbb{E}\|\tilde{V}_i - \tilde{V}_j\|^2\mathbb{1}(\zeta(i) = j)$$

$$\le (\tfrac{\delta_c(i)}{\delta_c(k)} + \tfrac{\delta_b(i)}{\delta_c(k)}(\tilde{n} - \tilde{f} - 1))2d\sigma^2$$

$$\le (\tfrac{\tilde{n}-\tilde{f}-2}{\tilde{n}-2\tilde{f}-2} + \tfrac{\tilde{f}}{\tilde{n}-2\tilde{f}-2}(\tilde{n} - \tilde{f} - 1))2d\sigma^2$$

Putting everything together we have

$$\|\mathbb{E}\text{KRUM} - \boldsymbol{g}\|^2 \le (\tilde{n} - \tilde{f})2d\sigma^2 + \tilde{f} \cdot (\tfrac{\tilde{n}-\tilde{f}-2}{\tilde{n}-2\tilde{f}-2} + \tfrac{\tilde{f}}{\tilde{n}-2\tilde{f}-2}(\tilde{n} - \tilde{f} - 1))2d\sigma^2$$

$$\le \eta^2(\tilde{n}, \tilde{f})d\sigma^2$$

By assumption $\eta^2(\tilde{n}, \tilde{f})d\sigma^2 < \|\boldsymbol{g}\|$, we know

$$\langle\mathbb{E}\text{KRUM}, \boldsymbol{g}\rangle \ge (\|\boldsymbol{g}\| - \eta(\tilde{n}, \tilde{f}) \cdot \sqrt{\tilde{f}} \cdot \sigma) \cdot \|\boldsymbol{g}\| = (1 - \sin\alpha) \cdot \|\boldsymbol{g}\|^2 \tag{14}$$

To sum up, (i) of Byzantine resilience holds. Now we focus on (ii),

$$\mathbb{E}\|\text{KRUM}\|^r = \sum_{\text{correct } i}\mathbb{E}\|\tilde{V}_i\|^2\mathbb{1}(i_* = i) + \sum_{\text{byz } k}\mathbb{E}\|\tilde{B}_k\|^2\mathbb{1}(i_* = k)$$

We think of $G$ as the estimator of gradients among good workers. More specifically, $G$ is a random variable which uniformly samples one worker among good workers and samples one gradient from the worker. Then $P(G = \boldsymbol{v}) = \tfrac{1}{n-f}\sum_{i=1}^{n-f}P(V_i = \boldsymbol{v})$, and

$$\mathbb{E}\|G\|^r = \sum_{\boldsymbol{v}}P(G = \boldsymbol{v})\|\boldsymbol{v}\|^r = \tfrac{1}{n-f}\sum_{i=1}^{n-f}\sum_{\boldsymbol{v}}P(V_i = \boldsymbol{v})\|\boldsymbol{v}\|^r = \tfrac{1}{n-f}\sum_{i=1}^{n-f}\mathbb{E}\|V_i\|^r \tag{15}$$

Thus

$$\sum_{\text{correct } i}\mathbb{E}\|\tilde{V}_i\|^r \le \sum_{i=1}^{n-f}\mathbb{E}\|V_i\|^r = (\tilde{n} - \tilde{f})\mathbb{E}\|G\|^r \tag{16}$$

We have

$$\mathbb{E}\|\text{KRUM}\|^r \le (\tilde{n} - \tilde{f})\mathbb{E}\|G\|^r + \sum_{\text{byz } k}\mathbb{E}\|\tilde{B}_k\|^2\mathbb{1}(i_* = k)$$

Denoting $C$ a generic constant, when $i_* = k$, we have for all correct indices $i$

$$\|\tilde{B}_k - \tfrac{1}{\delta_c(k)}\sum_{k\to\text{correct } j}\tilde{V}_j\| \le \sqrt{\tfrac{1}{\delta_c(k)}\sum_{i\to\text{correct}j}\|\tilde{V}_i - \tilde{V}_j\|^2 + \tfrac{\delta_b(i)}{\delta_c(k)}\|\tilde{V}_i - \tilde{V}_{\zeta(i)}\|^2}$$

$$\le C \cdot (\sqrt{\tfrac{1}{\delta_c(k)}}\sum_{i\to\text{correct}j}\|\tilde{V}_i - \tilde{V}_j\| + \sqrt{\tfrac{\delta_b(i)}{\delta_c(k)}})$$

$$\le C \cdot \sum_{\text{correct } j}\|\tilde{V}_j\|$$

Now

$$\|\tilde{B}_k\| \le \|\tilde{B}_k - \frac{1}{\delta_c(k)}\sum_{k\to\text{correct } j}\tilde{V}_j\| + \|\frac{1}{\delta_c(k)}\sum_{k\to\text{correct } j}\tilde{V}_j\|$$

$$\le C\sum_{\text{correct } j}\|\tilde{V}_j\|$$

$$\|\tilde{B}_k\|^r \le C\sum_{r_1+\cdots+r_{\tilde{n}-\tilde{f}}=r}\|\tilde{V}_1\|^{r_1}\cdots\|\tilde{V}_{\tilde{n}-\tilde{f}}\|^{r_{\tilde{n}-\tilde{f}}}$$

Take expectation to both sides and apply generalized Young's inequality

$$a_1^{p_1}\cdots a_m^{p_m} \le p_1a_1 + \cdots + p_ma_m \tag{17}$$

where $p_1 + \cdots + p_m = 1$, we have

$$\mathbb{E}\|\tilde{B}_k\|^r \le C\sum_{r_1+\cdots+r_{\tilde{n}-\tilde{f}}=r}(\tfrac{r_1}{r}\mathbb{E}\|\tilde{V}_1\|^r + \cdots + \tfrac{r_{\tilde{n}-\tilde{f}}}{r}\mathbb{E}\|\tilde{V}_{\tilde{n}-\tilde{f}}\|^r) \le C\mathbb{E}\|G\|^r \tag{18}$$

the last inequality comes from (16). Thus we have proven property (ii) of $(\alpha, \tilde{f})$-Byzantine resilience. $\square$

A.1    ALTERNATIVE CONVERGENCE PROOF OF KRUM WITH RESAMPLING

For completeness we also provide an alternative proof for a slight variation of the Byzantine Resilience definition, under the same algorithm of KRUM with resampling.

**Definition B** (($\alpha, f$)-Byzantine Resilience Alternative.). *Let $0 \leq \alpha < \pi/2$ be any angular value, and any integer $0 \leq f \leq n$. Let $V_1, \ldots, V_n$ be any identically distributed random vectors in $\mathbb{R}^d$, $V_i \sim G$, with $\mathbb{E}\, G = \boldsymbol{g}$. Let $B_1, \ldots, B_f$ be any Byzantine vectors in $\mathbb{R}^d$, possibly dependent on the $V_i$'s. An aggregation rule $F$ is said to be ($\alpha, f$)-Byzantine resilient if, for any $1 \leq j_1 < \cdots < j_f \leq n$, then*

$$F = F(V_1, \ldots, \underbrace{B_1}_{j_1}, \ldots, \underbrace{B_f}_{j_f}, \ldots V_n)$$

*satisfies (i) $\langle \mathbb{E}\, F, \boldsymbol{g} \rangle \geq (1 - \sin \alpha) \cdot \|g\|^2 > 0$ and (ii) for $r = 2, 3, 4$, $\mathbb{E}\, \|F\|^r \leq \mathbb{E}\, \|G\|^r$.*

Again, similarly as for the above Theorem II we can obtain almost sure convergence analogous to (Blanchard et al., 2017, Proposition 2)

**Theorem III** (Resampling KRUM, Alternative). *We assume that (i) the cost function $\mathcal{L}$ is three times differentiable with continuous derivatives and non-negative $\mathcal{L}(\boldsymbol{x}) \geq 0$; (ii) the learning rates satisfy $\sum_t \gamma_t = \infty$ and $\sum_t \gamma_t^2 < \infty$; (iii) the gradient estimator satisfies $\mathbb{E}\, G(\boldsymbol{x}, \boldsymbol{\xi}) = \nabla\mathcal{L}(\boldsymbol{x})$ and $\forall r \in \{2, 3, 4\}$, $\mathbb{E}\, \|G(\boldsymbol{x}, \boldsymbol{\xi})\|^r \leq A_r + B_r\|\boldsymbol{x}\|^r$ for some constants $A_r$, $B_r$; (iv) there exists a constant $0 \leq \alpha < \pi/2$ such that for all $\boldsymbol{x}$ we have $\eta(T, sf) \cdot \sqrt{d} \cdot \sigma(\boldsymbol{x}) \leq \|\nabla\mathcal{L}(\boldsymbol{x})\| \cdot \sin \alpha$; (v) finally, beyond a certain horizon, $\|\boldsymbol{x}\|^2 \geq D$, there exist $\varepsilon > 0$ and $0 \leq \beta < \pi/2 - \alpha$ such that $\|\nabla\mathcal{L}(\boldsymbol{x})\| \geq \varepsilon > 0$ and $\langle \boldsymbol{x}, \nabla\mathcal{L}(\boldsymbol{x}) \rangle \geq \cos \beta \|\boldsymbol{x}\| \cdot \|\nabla\mathcal{L}(\boldsymbol{x})\|$. If $s > 1$ and $2sf + 3 \leq n$, then*

- KRUM ∘*Resampling is ($\alpha, sf$)-Byzantine resilient as in Definition B where $0 \leq \alpha < \pi/2$ is defined by*

$$\sin \alpha = \frac{\eta(T, sf) \cdot \sqrt{d} \cdot \sigma}{\|\nabla\mathcal{L}(\boldsymbol{x})\|}, \quad \eta(n, f) := \sqrt{2\left(n - f + \frac{f \cdot (n-f-2) + f^2 \cdot (n-f-1)}{n - 2f - 2}\right)} \quad (19)$$

- *the sequence of gradients $\nabla\mathcal{L}(\boldsymbol{x}_t)$ converges almost surely to zero.*

*Proof.* A key difference between Definition B and (Blanchard et al., 2017, Definition 1) is that Definition B removes the *independence* requirement for the input good gradients $\{V_i\}_{i=1}^{n-f}$. In (Blanchard et al., 2017), there are two propositions: (Blanchard et al., 2017, Proposition 1) proves that KRUM satisfy (Blanchard et al., 2017, Definition 1), and (Blanchard et al., 2017, Proposition 2) shows almost surely convergence of the gradient $\nabla\mathcal{L}$.

In our case, we only need to show that KRUM ∘Resampling satisfy Definition B because the convergence of $\nabla\mathcal{L}$ is identical to the proof of (Blanchard et al., 2017, Proposition 2). Since the gradients after resampling is identically distributed according to Proposition I, we can keep all of the proofs of (Blanchard et al., 2017, Proposition 1) except for the last inequality where the independence is used.

$$\|B_k\|^r \leq C \sum_{r = r_1 + \cdots + r_{n-f}} \|V_1\|^{r_1} \cdots \|V_{n-f}\|^{r_{n-f}} \quad (20)$$

By applying expectation to both sides of the inequality and the independence of $\{V_i\}_{i=1}^{n-f}$, they conclude that

$$\mathbb{E}\|B_k\|^r \leq C \sum_{r = r_1 + \cdots + r_{n-f}} \mathbb{E}\|V_1\|^{r_1} \cdots \mathbb{E}\|V_{n-f}\|^{r_{n-f}} = C \sum_{r = r_1 + \cdots + r_{n-f}} \|G\|^{r_1} \cdots \|G\|^{r_{n-f}} \quad (21)$$

We can immediately show that $\mathbb{E}\|B_k\|^r \leq C\|G\|^r$ with the help of a general form of Young's inequality. We prove this using the standard Young's inequality. The Young's inequality states that for $p > 1, q > 1, \frac{1}{p} + \frac{1}{q} = 1, a, b \geq 0$, we have

$$ab \leq \frac{a^p}{p} + \frac{b^q}{q} \quad (22)$$

Let $a = \|V_1\|^{r_1}, b = \|V_2\|^{r_2} \cdots \|V_{n-f}\|^{r_{n-f}}$, and $p = \frac{r}{r_1}, q = \frac{r}{r - r_1}$, we apply Young's inequality

$$\|V_1\|^{r_1} \cdots \|V_{n-f}\|^{r_{n-f}} \leq \frac{r_1}{r}\|V_1\|^r + \frac{r - r_1}{r}\|V_2\|^{\frac{rr_2}{r - r_1}} \cdots \|V_{n-f}\|^{\frac{rr_{n-f}}{r - r_1}}$$

We can apply Young's inequality agin for the second term of right hand side. Let $a = \|V_2\|^{\frac{rr_2}{r-r_1}}$, $b = \|V_3\|^{\frac{rr_3}{r-r_1}} \cdots \|V_{n-f}\|^{\frac{rr_{n-f}}{r-r_1}}$, and $p = \frac{r-r_1}{r_2}, q = \frac{r-r_1}{r-r_1-r_2}$, we apply Young's inequality

$$\|V_1\|^{r_1} \cdots \|V_{n-f}\|^{r_{n-f}} \leq \frac{r_1}{r}\|V_1\|^r + \frac{r_2}{r}\|V_2\|^r + \frac{r-r_1-r_2}{r}\|V_3\|^{\frac{rr_3}{r-r_1-r_2}} \cdots \|V_{n-f}\|^{\frac{rr_{n-f}}{r-r_1-r_2}} \leq \sum_{i=1}^{n-f} \frac{r_i}{r}\|V_i\|^r$$

where the second inequality results from recursively applying Young's inequality. Then we apply expection to both sizes of the inequality gives

$$\mathbb{E}\|V_1\|^{r_1} \cdots \|V_{n-f}\|^{r_{n-f}} \leq \sum_{i=1}^{n-f} \frac{r_i}{r}\mathbb{E}\|V_i\|^r = \|G\|^r$$

Then applying expectation to Equation (20) gives

$$\mathbb{E}\|B_k\|^r \leq C \sum_{r=r_1+\cdots+r_{n-f}} \mathbb{E}\|V_1\|^{r_1} \cdots \|V_{n-f}\|^{r_{n-f}} = C\|G\|^r \tag{23}$$

where $C$ is a general coefficient. $\qquad\square$

## B  CONVERGENCE OF BYZANTINE RESILIENT SGD

In this section, we obtain a finite-time convergence guarantee for any algorithm which satisfies $(\alpha, f)$-Byzantine resilience. As far as we are aware, this is the first convergence result for KRUM.

From Theorem II, we know that Suppose that the following condition holds for any iterations $k$ for some constants $\delta \in (0, 1]$ and $\beta \geq 0$ such that

$$\|\mathbb{E} F - g\|^2 \leq (1-\delta)\|g\|^2, \text{ and } \mathbb{E}\|F - \mathbb{E} F\|^2 \leq \beta^2, \tag{24}$$

where $F$ is the output of the robust-aggregation algorithm and $g := \nabla f(x_k)$. This first condition bounds the bias wheras the second part bounds the variance of $F$.

**Convergence.**

**Theorem IV.** *Given any biased stochastic estimator $F$ satisfying* (24)*, the following holds for the update $x_{k+1} = x_k - \eta F$ and an L-smooth potentially non-convex $f(x)$ lower-bounded by $f^\star$:*

$$\frac{1}{K}\sum_{k=0}^{K-1}\mathbb{E}\|\nabla f(x_k)\|^2 \leq \frac{2L(f(x_0) - f^\star)}{\delta K} + \sqrt{\frac{8L\beta^2(f(x_0) - f^\star)}{\delta^2 K}}$$

*Proof.* The following holds for any $\eta \leq \frac{1}{L}$:

$$\mathbb{E}_k[f(x_{k+1})] \leq f(x_k) - \mathbb{E}\,\eta\langle g, F\rangle + \frac{\eta^2 L}{2}\|F\|^2$$

$$\leq f(x_k) - \mathbb{E}\,\eta\langle g, F\rangle + \frac{\eta^2 L}{2}\|\mathbb{E} F\|^2 + \frac{\eta^2\beta^2 L}{2}$$

$$\leq f(x_k) - \frac{\eta}{2}\|g\|^2 + \frac{\eta}{2}\|\mathbb{E} F - g\|^2 + \frac{\eta^2\beta^2 L}{2}$$

$$n \leq f(x_k) - \frac{\eta\delta}{2}\|g\|^2 + \frac{\eta^2\beta^2 L}{2}.$$

The first inequality uses the smoothness of $f$, the second separated and bounded the variance of $F$ by $\beta^2$, third inequality follows from rearranging terms and using $\eta \leq \frac{1}{L}$, and the final inequality used our bound on the bias of $F$. Now rearranging the terms and averaging over $k$ gives:

$$\frac{1}{K}\sum_{k=0}^{K-1}\mathbb{E}\|g_k\|^2 \leq \frac{2(f(x_0) - f(x_k))}{\eta\delta K} + \frac{\eta L\beta^2}{\delta} \leq \frac{2(f(x_0) - f^\star)}{\eta\delta K} + \frac{\eta L\beta^2}{\delta}.$$

Picking $\eta = \min(\frac{1}{L}, \sqrt{\frac{2(f(x_0)-f^\star)}{LK\beta^2}})$ gives the desired rate. $\qquad\square$

**Relation to other conditions.** If $F$ is an unbiased stochastic gradient $\mathbb{E}[F \mid x_k] = g$ and variance $\sigma^2$, then (24) holds with $\delta = 1$ and $\beta = \sigma$. In this case, Thm. IV recovers the standard convergence of SGD for smooth non-convex functions.

If instead, we have $\langle \mathbb{E} F, g \rangle \geq (1 - \sin(\alpha)) \|g\|^2$, and $\mathbb{E}\|F\|^2 \leq \|g\|^2 \leq G^2$,

$$
\begin{aligned}
\|\mathbb{E} F - g\|^2 =& \|g\|^2 - 2\langle \mathbb{E}[F], g \rangle + \|\mathbb{E}[F]\|^2 \\
\leq& \|g\|^2 - 2\langle \mathbb{E}[F], g \rangle + \mathbb{E}\|F\|^2 \\
\leq& 2\sin(\alpha)\|g\|^2 .
\end{aligned}
$$

Thus, in this case (24) holds with $\delta = 1 - 2\sin(\alpha)$, and $\beta = G$. Here, we only get convergence if $\sin(\alpha) \leq \frac{1}{2}$. Further, the assumption that $\mathbb{E}\|F\|^2 \leq \|g\|^2 \leq G^2$ is extremely strong.

## C   RESAMPLING WITH REPLACEMENT FOR ROBUST LEARNING

---
**Algorithm 3** Resampling with replacement

---
**Input:** $\{\boldsymbol{g}_i : i \in [n]\}, T, s$
**for** $t = 1, \dots, T$ **do**
    Uniformly sample $\{\boldsymbol{g}_{j_i}\}_{i=1}^{s}$ from $\{\boldsymbol{g}_i : i \in [n]\}$
    Compute average $\bar{\boldsymbol{g}}_t := \frac{1}{s} \sum_{i=1}^{s} \boldsymbol{g}_{j_i}$
Return $\{\bar{\boldsymbol{g}}_t : t \in [T]\}, \{j_i^t : t \in [T], i \in [s]\}$

---

Contrary to resampling without replacement which has an explicit bound for malicious gradients after resampling, its with-replacement version has no guarantee for the honest majority.

**Lemma 3.** *Given a set of vectors $\{\boldsymbol{g}_i : i \in [n]\}$ of any distributions, the output of Algorithm 3 $\{\bar{\boldsymbol{g}}_t : t \in [T]\}$ are identically distributed with expectation*

$$
\mathbb{E}\bar{\boldsymbol{g}}_t = \frac{1}{n} \sum_{i=1}^{n} \mathbb{E}\boldsymbol{g}_i \qquad \forall\, t \in [T] \tag{25}
$$

*If $f$ of the inputs are Byzantine, then a subset $\mathcal{S} \subset \{\bar{\boldsymbol{g}}_t : t \in [T]\}$ are not faulty. The cardinality of $\mathcal{S}$ is $\mathbb{E}|\mathcal{S}| = \lfloor T(1 - f/n)^s \rfloor$ and the gradients are identically distributed with expectation*

$$
\mathbb{E}\bar{\boldsymbol{g}} = \frac{1}{|\boldsymbol{good}|} \sum_{i \in \boldsymbol{good}} \mathbb{E}\boldsymbol{g}_i \qquad \forall\, \bar{\boldsymbol{g}} \in \mathcal{S} \tag{26}
$$

*Proof.* $\mathbb{E}\bar{\boldsymbol{g}}_t = \mathbb{E}\frac{1}{s} \sum_{i=1}^{s} \boldsymbol{g}_i = \frac{1}{s} \sum_{i=1}^{s} \mathbb{E}\boldsymbol{g}_i$.

There are $(1 - f/n)$ chance that a sampled gradient $\boldsymbol{g}_{j_i}$ is good and $(1 - f/n)^s$ chance that all of the $s$ sampled gradients are good. Repeat the resampling for $T$ times gives $\mathbb{E}|\mathcal{S}| = \lfloor T(1 - f/n)^s \rfloor$. $\qquad\square$

**Remark 4.** *The $\bar{\boldsymbol{g}}_t$ has same expectation as minibatch sgd.*

We denote the expected number of Byzantine gradients in $\{\bar{\boldsymbol{g}}_t : t \in [T]\}$ as follows:

$$
\tilde{f} := T - \mathbb{E}|\mathcal{S}| = T(1 - \lfloor (1 - f/n)^s \rfloor) \tag{27}
$$

Since the vectors in $\mathcal{S}$ are identically distributed, we can apply robust aggregation rule $\mathcal{A}$, like Multik-KRUM, to $\{\bar{\boldsymbol{g}}_t : t \in [T]\}$. The convergence of Algorithm 1 for Multi-KRUM is stated below.

**Theorem V** (Resampled KRUM). *Assume the dataset is decentralized stored. Assume other conditions in (Blanchard et al., 2017, Prop. 1 & 2) hold true. $K$ is the number of aggregations. If $2\tilde{f} + 2 < T$, then with a probability $p^K$, KRUM∘RSWR is Byzantine robust and the sequence of gradients almost surely converges to zero. The $p$ is defined as*

$$
p := \sum_{i=0}^{\lfloor T/2 \rfloor} q^{\lceil T/2 \rceil + i} (1 - q)^{\lfloor T/2 \rfloor - i} \tag{28}
$$

*where $q = (1 - f/n)^s$*

*Proof.* The probability of $\bar{\boldsymbol{g}}_t$ is good is $q = (1 - f/n)^s$. The probability of non-faulty majority after RSWR is thus

$$p := \sum_{i=0}^{\lfloor T/2 \rfloor} q^{\lceil T/2 \rceil + i} (1 - q)^{\lfloor T/2 \rfloor - i}$$

The output of RSWR are identically distributed by Lemma 3. Thus we know that the robust aggregation rule converge with probability $p^K$. □

**Remark 5.** *Consider the range of $T$, $s$, $f$. Since Theorem V requires $2\tilde{f} + 2 < T$, we need $s < \frac{\log(1/2 + 1/T)}{\log(1 - f/n)}$. Furthermore, $s \geq 1$ is lower bounded, $T > \frac{2n}{n - 2f}$ which only requires $2f < n$.*

**Remark 6.** *Notice that the cardinality $|\mathcal{S}|$ is stochastic which means it is possible to have a faulty majority in any round. if the Byzantine gradients are very large, like gaussian attack, the model diverges as soon as one Byzantine gradient is selected. On the other hand, in the experiment section, we show that for many attacks (labelflipping, bitflipping, etc.) the error introduced by faulty majority rounds are amortized overtime. To fix this issue, the server can normalize all the gradient by their norm such that a faulty majority round would not lead to catestrophic consequences.*

# D ADDITIONAL EXPERIMENTS

## D.1 FASHION-MNIST

In this subsection, we demonstrate that our algorithm also works on modern dataset like Fashion-MNISTXiao et al. (2017). Since the Fashion-MNIST is designed to be a drop-in replacement of MNIST, we conduct experiments on Fashion-MNIST with same setups as Figure 3. The results are presented in Figure 4.

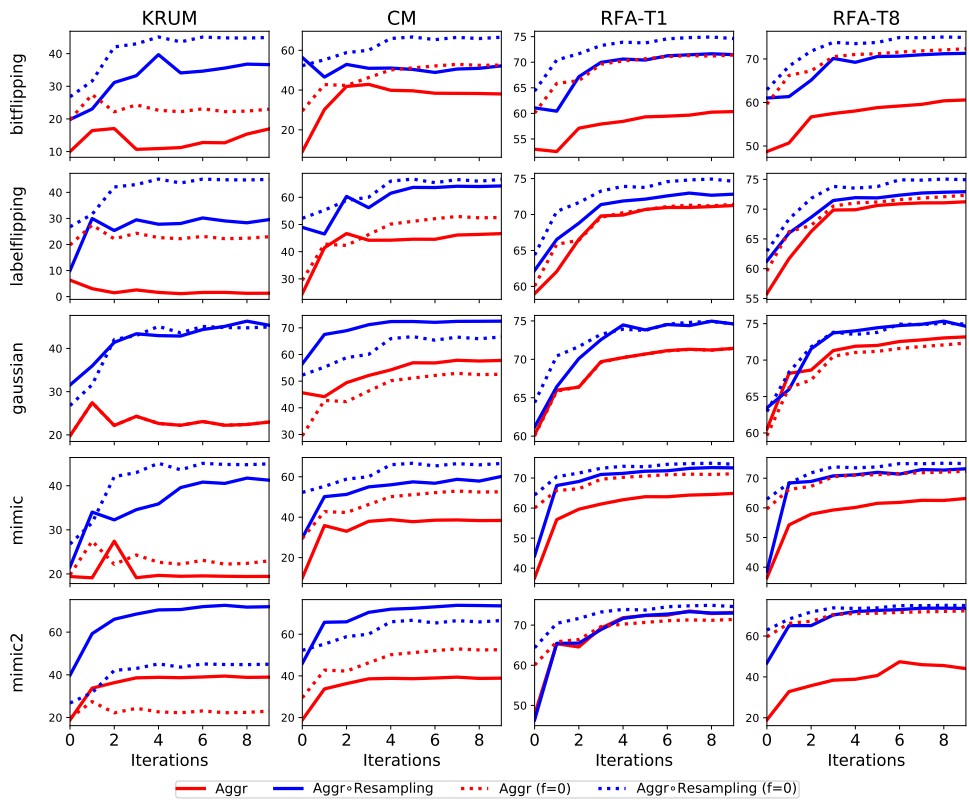

Figure 4: Test accuracies of KRUM, CM, RFA under 5 kinds of attacks (and without attack) on non-iid Fashion-MNIST datasets. There are 12 workers and 2 of them are Byzantine according to each attack row. Columns show each aggregation rule applied without (red) and with resampling (blue). Dotted lines for comparison are showing the same method without any Byzantine workers ($f = 0$). For RFA, T1, T8 refers to the number of inner iterations of Weiszfeld's algorithm.

## D.2    CIFAR-10

We run experiments on CIFAR-10 (Krizhevsky et al., 2009) with ResNet-20 (He et al., 2015). We train our model on 12 nodes which includes 2 Byzantine attacker with mimic attack. We use KRUM as the aggregation rule for demonstration. We choose learning rate to be $0.1$ and batch size per node to be $120$. The CIFAR-10 dataset is splitted across good nodes such that 50% of samples on each good node has same distribution as overall distribution, the rest 50% are samples from a single class which is different among good workers. We present our results in Figure 5.

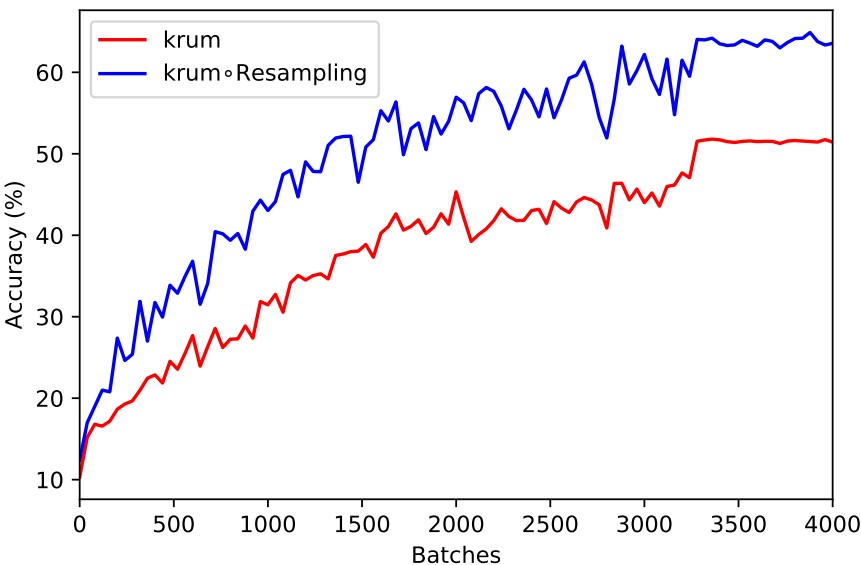

Figure 5: Compre KRUM and KRUM ∘Resampling for training ResNet-20 on CIFAR-10 dataset. There are 10 good workers and 2 Byzantine workers.

Note that the accuracy in this example is lower than the normal setting because krum may bias towards certain nodes. Besides, the batch norm maybe influenced by the heterogenous distribution of data.

## D.3 RESAMPLING OR FIXED GROUPING

In (Chen et al., 2017), workers are grouped at the beginning of training, and they are trained on i.i.d datasets. In contrast, resampling is performed every round, and applies to non-iid datasets. If a Byzantine worker can predict the random bits on server, resampling becomes grouping in each round which is still stronger than (Chen et al., 2017).

In Figure 7, we compare KRUM ∘resampling with vanilla KRUM and KRUM with fixed grouping. As we can see, the fixed grouping has better accuracy than vanilla KRUM, but weaker than resampling as we expected.

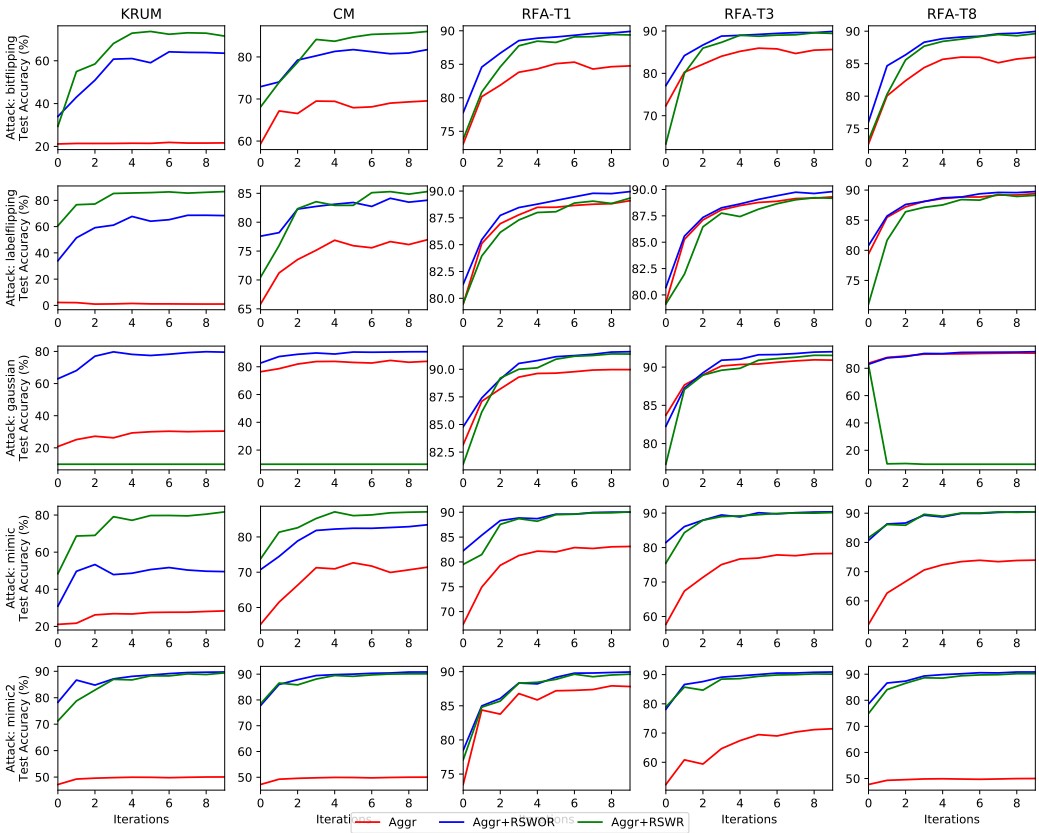

Figure 6: Comparing 3 aggregation rules under 5 kinds of attacks on non-iid datasets. There are 10 workers and 2 of them are Byzantine. In the grid of experiments, same aggregation rules are used in the same column and same attacks are applied to the same row. The aggregation rules are KRUM (Blanchard et al., 2017), CM (Yin et al., 2018a), RFA (Pillutla et al., 2019). The RFA-T1, T3, T8 refers to the number of inner iterations.

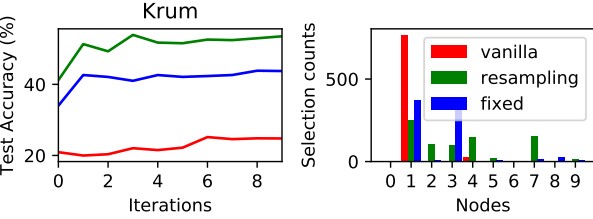

Figure 7: Comparison with no resampling, and fixed grouping for KRUM on non-i.i.d datasets.

### D.4 RESAMPLING HYPERPARAMETER

The resampling hyperparameter $s$ controls the variance reduction, as has been stated in Proposition I. In Figure 8, we compare the performance of no resampling and resampling with $s = 2, 3, 4$ on heterogenous MNIST dataset. There are 10 workers in total and no Byzantine workers. Each experiment has been run for 5 times.

As we can see from Figure 8, higher $s$ leads to faster convergence. It matches with the Proposition I that higher $s$ leads to greater variance reduction.

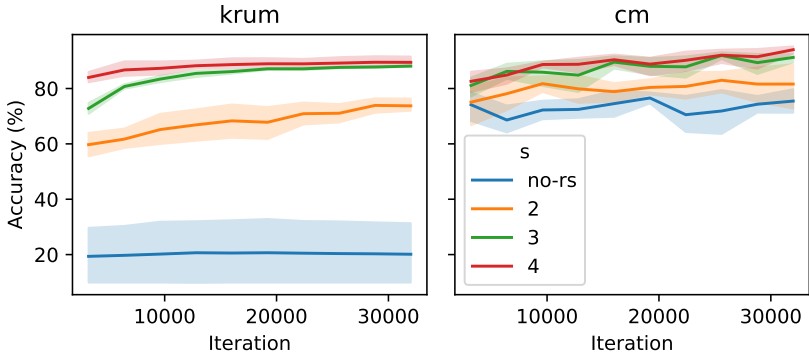

Figure 8: Compare no resampling with $s = 2, 3, 4$ on MNIST data. There are 10 workers and 0 Byzantine worker.

