# OpenReview forum: "Byzantine-Robust Learning on Heterogeneous Datasets via Resampling"
_ICLR.cc/2021/Conference — Reject_

### Official Review · AnonReviewer2 · 2020-10-25
**A good attempt towards robust federated learning**

**Rating:** 6
**Confidence:** 4

**Review:**

Paper summary:

The paper studies Byzantine robustness in the context of distributed learning from heterogeneous datasets. This problem has been widely studied previously, but under the additional assumption that the data of the good workers is i.i.d.. The authors give examples of situations and poisoning attacks with which current defences designed for the i.i.d. situation can be overcome.  They also propose a simple resampling scheme that can be used as a preprocessing step before applying any standard robust aggregator from the i.i.d. literature. They provide theoretical guarantees for their resampling scheme when used together with KRUM. The also test their algorithm against the i.i.d. baselines and multiple attacks.

##########################################################################

Pros:

- The paper is well-written and motivated and addresses an important problem. Indeed, in the context of distributed and federated learning, where privacy and communication costs are of particular importance, the problem of corrupted or adversarial updates becomes very relevant.
- The gap between existing defences and the non-i.i.d. setup is well-motivated, with various examples and attacks proposed in the paper.
- The proposed resampling scheme is simple and easy to incorporate into previously proposed algorithms. It is also amendable to theoretical analysis, as shown in the paper.
- The experimental evaluation does show clear improvements in many cases when the resampling scheme is used.

##########################################################################

Cons/suggestions:

- While heterogeneous data is allowed, the assumption on the top of page 2 that the variance between the gradients of the local datasets is bounded is somewhat unintuitive to me. It seems like this is not just the usual assumption that the variance of the stochastic updates is bounded, but it's also a measure of how non-i.i.d. the datasets are. Is there an intuitive explanation of what such a bound means in terms of how the distributions of the clients vary?
-  While the resampling method is indeed relatively simple and easy to understand, I think that the paper will benefit from a discussion on why it actually helps. For example Remark 1 could be followed by some examples of situations when mixing the gradients in advance will help or alternatively some comparison to other fields where such resampling approaches were found useful. Similarly, it would be interesting to see a discussion of how much and in what respect does the proof of Krum change once dependence between the vectors is introduced.
- For the experiments, it would be nice to see plots for different values of $s$, not only $s = 2$, so as to see if the method effectively trades-off convergence speed with robustness, as claimed in Remark 1.
- It will also be nice to get more information about the data splits between the clients, as well as experiments with multiple repeats and error bars and/or multiple amount of heterogeneity introduced, so that the significance of the improvements can be easier to evaluate.

##########################################################################

Minor points:

- In Algorithm 1 the inputs passed on to Algorithm 2 are different that the inputs received in Algorithm 2. It would be nice to make these consistent.
- In Proposition 1 it is assumed that the input is a set of gradients. However, in the second bullet point $f$ gradients are assumed to be Byzantine. It's a bit unclear what this means given that we are just talking about $n$ arbitrary vectors. I think I understand what is meant, but it will be nice to clarify.


##########################################################################

Review summary:

The paper shows that a simple, but effective resampling scheme can be used to transfer i.i.d. Byzantine-robust algorithms to a non-i.i.d. setup. The paper is generally well-written and provides some theoretical and experimental evidence to support the proposed method. Therefore, I recommend acceptance, together with a few suggestions for improving the discussion and the experimental section.

---

> ### Author Response · Authors · 2020-11-15
> **Reply R2 - Bounded variance bounds non-iid-ness and is a standard assumption.**
>
> We thank the reviewer for their enthusiasm as well as their detailed and helpful suggestions. We will address the main concerns below.
>
> > Need for the assumption of bounded variance across good workers.
>
> This assumption indeed bounds the amount of non-iidness between the workers’ datasets. If they can be arbitrarily different, then it is impossible for the server to make out whether some worker who is sending “very weird” gradients is actually acting adversarially or just has “very weird” data. Hence it is necessary to bound the amount of non-iid-ness among the good workers. Similar assumptions are standard in the analysis of Byzantine-robust learning (Data and Diggavi, 2020; Chen et al., 2020) as well as vanilla federated learning [1].
>
> > Meaning of bounded variance in terms of data distribution.
>
> Since the gradients across different workers computed at the same parameters $x$ differs only due to the data distribution, it is intuitive that the larger is the difference between the data distributions, the larger is the variance in the gradients. Making this relation precise is non-trivial and depends on the range of parameters (hypothesis space) being considered. Intuitively, $\bar \sigma^2$ can be seen as a generalization of a notion of *discrepancy* which is defined over the gradients instead of function values. Sec. 2.2 of [1] for additional discussion of this assumption.
>
>
> > Experiments with varying s, repetitions with different seeds, and varying heterogeneity.
>
> In all non-iid experiments, we split the data by sorting their labels and sequentially assign samples to workers such that the data distributions are extremely heterogeneous. We have added the experiment with varying $s$ in Appendix D.4. The results show that higher $s$ leads to faster convergence which matches our proposition. We are currently running the experiments and we will update the results here in this thread.
>
> [1] Bayoumi et al. "Tighter theory for local SGD on identical and heterogeneous data." AISTATS 2020.

---

> > ### Comment · AnonReviewer2 · 2020-11-22
> > **Thank you for the clarifications.**
> >
> > Thank you for the clarifications. I realize that assuming a bound on the non-i.i.d.-ness of the datasets is important for the analysis. In this sense I agree that the assumed bound on the gap between the gradients, although somewhat hard to interpret, is a good start.
> >
> > Looking forward to seeing the updated experiments.

---

> > > ### Author Response · Authors · 2020-11-23
> > > **We have updated the experiments with varying $s$, more experiments are running.**
> > >
> > > We thank Reviewer 2 for the reply. We have updated the manuscript with additional experiments on varying $s$ in Appendix D.4. We will add more experiments with different random seeds for Figure 3.

---

> > > ### Author Response · Authors · 2020-11-23
> > > **We have updated the experiments in Figure 3**
> > >
> > > We have updated Figure 3 in the main text by running each experiment 5 times. Note that we made a few minor changes in the experimental setups and plot styles (not the algorithm) but not the resampling algorithm and aggregation rules. The conclusions draw from Figure 3 remain the same.

---

### Official Review · AnonReviewer1 · 2020-10-26
**Simple idea with good performance in practice and easy to implement, good paper**

**Rating:** 7
**Confidence:** 3

**Review:**

In this paper, the authors propose the resampling mechanism to improve the performance of existing robust statistics method for Byzantine-robust distributed SGD. Both theoretical and empirical analysis are provided. In overall, the proposed method is technically sound, and works well in practice.

Here is some detailed comments:

1. According to my understanding, the resampling mechanism is actually trying to convert non-iid gradients into iid ones. If an averaged random subsample of gradients contains only good gradients, then sampling with replacement makes it an unbiased estimator. If the averaged random subsample contains Byzantine gradients, then it is corrupted anyway and we don't care it's biased or not. Thus, for the averaged subsamples containing no Byzantine gradients, they actually become iid. Is my understanding correct?

2. The idea is simple and easy to implement upon all the previous work. Some people may think this work is incremental, but I think the simplicity of the proposed method is an advantage.

3. It will be better if the authors also provide the convergence analysis of resampling with CM. For Krum, note that larger n (number of workers) actually results in worse convergence in Theorem II, which questions whether we should use distributed training.

---

> ### Author Response · Authors · 2020-11-15
> **Reply R1- We agree that resampling is a simple and intuitive approach**
>
> We thank the reviewer for their enthusiastic response and for their detailed comments. We address them below.
>
> 1. Yes, this is exactly our intuition.
> 2. We wholeheartedly agree and believe simplicity should be prized.
> 3. CM indeed has better convergence analysis results than Krum, though with the additional assumption of bounded skewness and a much more elaborate proof. Given the added complexity of the proof of convergence of coordinate wise median, we feel that including it would add unnecessary complexity to the current work.

---

### Official Review · AnonReviewer4 · 2020-10-29
**This paper proposes an attack on distributed training with heterogeneous data distribution, along with a defense to it**

**Rating:** 5
**Confidence:** 3

**Review:**

Many existing researches on distributed deep learning works in byzantine robustness in a centralized PS setting under the assumption of i.i.d. data distribution on workers, e.g. KRUM. This paper presents a simple resampling scheme that adapts the existing robust algorithms to heterogeneous datasets (referred as KRUM-RS later in this review). It firstly proposes two new attacks under non-iid data distribution that fooled byzantine fault tolerant algorithms like KRUM, Coordinate Median (CM), and RFA (Normalized Median-NM or Geometric Median-GM). Then it proposes the RS algorithm and proved its theoretical convergence guarantee the of KRUM algorithm over non-iid data, and when the parameter server (PS) does not control the dataset distribution. Experiments showed the convergence guarantee over KRUM and even CM, NM/GM algorithm.

The proposed resampling algorithm is somewhat novel and can be applied to KRUM to guarantee its convergence. The paper is technically sound. The organization of this paper is OK, but it lacks clarifications on its intuition on the sampling method (why and how it improves other methods by sampling). The technical part is not self-contained. For example, in Def. A, what does it mean for r=2,3,4 that E(F) <= E(G)?  More clear explanation and definition should be present.


This paper seems to over-claim some of its results: for example, this paper claims that "We propose a simple new resampling step which can be used before any existing robust aggregation rule", but the theoretical analysis only applies to the rule from KRUM. How could one should the new resampling method work for any robust aggregation rule? There seems to be a large gap here.

---

> ### Author Response · Authors · 2020-11-15
> **Reply Reviewer 4 - Intuitively, resampling converts non-iid gradients into much more iid gradients**
>
> We thank the reviewer for their succinct summary of our work and their comments on our paper. We believe we address all concerns raised below and hope that the reviewer re-evaluates our paper in this light.
>
> > The intuition behind resampling is unclear.
>
> The main idea behind our defense is as follows: i) current schemes fail because gradients are non-iid, ii) after resampling, the resulting set of averaged gradients $\{\bar g_t : t \in [T ]\}$ are much more iid, iii) combining existing schemes with resampling enables them to work on non-iid data.
> We expand upon this intuition further in Section 4 and formalize it in Proposition 1.
>
> > Technical part is not self-contained
>
> We thank the reviewer for bringing this to our notice. Our priority was to keep the definitions and assumptions as close as possible to the original work Krum (Blanchard et al., 2017). By this, we wanted to demonstrate that our resampling scheme works in a *plug and play* manner, both experimentally and theoretically. We direct the reader to the original work of Krum (Blanchard et al., 2017) for the implications and justification for the particular choice of assumptions such as "$\mathbb{E}\|F\|^r \leq \mathbb{E}\|G\|^r$ for $r=2,3,4$".
>
> > seems to over-claim some of its results; results proved only for Krum
>
> Unfortunately, there does not exist a uniform framework of proof for the various aggregation schemes. The theoretical results in each of Krum, coordinate-wise median, and RFA rely on vastly differing assumptions and guarantees. E.g. Krum assumes Definition A, while the coordinate-wise median (Yin et al., 2018) assumes bounded skewness, and RFA (Pillutla et al., 2019) assumes none of these.
> Further, as we stated above, the proof in the case of RS-Krum is nearly identical to that of the original Krum with the additional use of Proposition 1. The proofs in the other cases can be adapted with similar ease to the non-iid setting when combined with resampling.
>
> Given the above, we decided that there is little advantage to reproving all the results above, and instead appeal to intuition that resampling makes the gradients much more iid, and so should benefit any scheme which requires iid gradients. Further, the experimental results clearly support our claim as resampling improves all three aggregation rules on numerous tasks and attacks.

---

### Decision · Program_Chairs · 2021-01-07
**Final Decision**

**Decision:**

Reject

**Comment:**

This paper presents an algorithm for distributed optimization in that aims to be "Byzantine-robust", in the sense that it learns successfully when some of the workers send arbitrary messages.  The goal in this work is to remain robust when each worker samples data from a different distribution.

While reviewers found the work interesting, issues about the theoretical development arose during the discussion period, and it appears that the paper cannot be accepted in its current form.

The most serious issue was with Proposition I, which appears to be incorrect.  In its putative proof, the authors write that each gradient is sampled at most $s$ times.  This naturally leads to the conclusion that, in Algorithm 2, when the Break statement is reached,  $g_{j_i}$ is not used to compute $\bar{g}_t$.

Given this interpretation of Algorithm 2, it seems that Proposition I cannot be true. For example, if $s=1$, once $t \geq n/2$, fairly often, gradients will be sampled that had previously been sampled. In such cases, would be zero, so that, on average, $\bar{g}_t$ would be biased toward zero in later rounds.

Their putative proof of Proposition I refers to a whole chapter of a statistics text. We couldn't find anything in that chapter that implies what they claim about Algorithm 2 (or that treats a sampling scheme like Algorithm 2 at all).

Throughout the paper, when the authors took expectations, it was not always clear what was random and what was fixed.  After some discussion, disagreement remained about how to interpret some of the assumptions.  This was true in particular about the assumption in the first displayed equation on page two.

---

> ### Author Response · Authors · 2021-01-15
> **Clarification on the correctness of the algorithm 2 and proposition 1**
>
> We would appreciate the time the reviewers and program chair spent on our paper, but we have to clarify that our algorithm 2 and proposition 1 are correct except that we forgot to delete the legacy condition ``If c[j_i] == s`` during our writing. The "If" is not in bold font like other statements. We meant to just put ``Break;`` at that line. The resampling with $s$-replacement are implemented correctly in the experiments. We address the issues mentioned in the final decision as follows.
>
> > This naturally leads to the conclusion that, in Algorithm 2, when the Break statement is reached $g_{j_i}$,  is not used to compute $\bar{g}_t$.
>
> About the technical details of Algorithm 2. There are 3 loops: T-loop, s-loop and a while loop. The ``Break`` statement jumps out of the while loop and gets a $j_i$ for the $i$. So the $g_{j_i}$ is used to compute $\bar{g}_t$.
>
> > For example, if $s=1$, once $t\ge n/2$, fairly often, gradients will be sampled that had previously been sampled. In such cases, would be zero, so that, on average, $\bar{g}_t$ would be biased toward zero in later rounds.
>
> When $s=$ and $t\ge n/2$, the  ``if c[j_i] < s then`` statement will reject previously sampled gradients. More generally, for any $s>0$ and $t<=n$, if a class ``c[j_i]`` has been sampled $s$ times, then $c[j_i] == s$ and we will not enter the if-clause and thus won't jump out of the while-loop. This is how we ensure each gradient is sampled at most $s$ times.
>
> Thus there will be no zero vectors and the gradients won't bias towards 0.
>
>
> > Their putative proof of Proposition I refers to a whole chapter of a statistics text. We couldn't find anything in that chapter that implies what they claim about Algorithm 2 (or that treats a sampling scheme like Algorithm 2 at all).
>
> We use the theorem B in Page 194 of the book (or page 213 of the pdf) by replacing N with sn. The intuition is that resampling with s-replacement is equivalent to replicating all gradients for $s$ times and perform simple random sampling(s) in it. For example, resampling with 1-replacement is simply shuffling (when T=n). Thus the proposition 1 is correct. (The page number we mentioned is based on a pdf version of the book we found online http://dase.ecnu.edu.cn/mgao/teaching/UStat_2018_Fall/mathematical+statistics+and+data+analysis.pdf)
>
>
> Finally, could you elaborate more on the ambiguity on the assumption in page 2? We could have explain all these issues if they are mentioned in the reviews or discussion phase.